

# Associations between psychometrically assessed life history strategy and daily behavior: data from the Electronically Activated Recorder (EAR)

Joseph H. Manson

Department of Anthropology, University of California, Los Angeles, Los Angeles, CA,
United States of America

## ABSTRACT

Life history theory has generated cogent, well-supported hypotheses about individual differences in human biodemographic traits (e.g., age at sexual maturity) and psychometric traits (e.g., conscientiousness), but little is known about how variation in life history strategy (LHS) is manifest in quotidian human behavior. Here I test predicted associations between the self-report Arizona Life History Battery and frequencies of 12 behaviors observed over 72 h in 91 US college students using the Electronically Activated Recorder (EAR), a method of gathering periodic brief audio recordings as participants go about their daily lives. Bayesian multi-level aggregated binomial regression analysis found no strong associations between ALHB scores and behavior frequencies. One behavior, presence at amusement venues (bars, concerts, sports events) was weakly positively associated with ALHB-assessed slow LHS, contrary to prediction. These results may represent a challenge to the ALHB's validity. However, it remains possible that situational influences on behavior, which were not measured in the present study, moderate the relationships between psychometrically-assessed LHS and quotidian behavior.

## INTRODUCTION

Life History Theory (LHT) describes and explains evolutionary processes that shape organisms' allocations of energy among the competing demands of growth, bodily maintenance, and reproduction (including courtship and parental investment) (*MacArthur & Wilson, 1967*; *Pianka, 1970*). On both phylogenetic and ontogenetic time scales, age-specific extrinsic mortality rates are a major driver of variation in Life History Strategy (LHS), such that higher extrinsic mortality rates select for a "faster" LHS, which prioritizes reproductive effort over somatic (i.e., growth and maintenance) effort, mating effort over parenting effort, and offspring quantity over offspring quality (*Charnov, 1993*; *Stearns, 1992*). Lower extrinsic mortality rates select for a "slower" LHS, which prioritizes the opposite energy allocations.

A large body of research, building on the seminal work of *Rushton (1985)* and *Belsky, Steinberg & Draper (1991)* has used LHT to generate and test hypotheses about human

Corresponding author
Joseph H. Manson,
jmanson@anthro.ucla.edu

individual and group differences. Three general approaches are distinguishable. First, research into the *biodemographic* aspects of LHS variation explores environmental influences on LH traits such as growth rates, age at first reproduction, and interbirth intervals. For example, in a multivariate analysis of a sample of small-scale societies, indicators of greater food availability were associated with faster and earlier child growth, but higher survival rates were associated with slower and later development (*Walker et al., 2006*). Among British neighborhoods, greater deprivation was associated with lower healthy life expectancy, earlier age at first birth, and shorter breastfeeding duration (*Nettle, 2010*). Second, researchers have examined whether, under laboratory conditions, LHT-related individual differences predict differences in responses to theoretically specified stimuli. For example, some findings indicate that low childhood socioeconomic status (cueing a faster LHS) interacts with mortality-risk primes to increase risk-proneness and reward delay discounting (*Griskevicius et al., 2011*; but see *Pepper et al., 2017*). Third, the *psychometric* approach to understanding LHS variation has posited theory-based cognitive and behavioral manifestations of LHS, and has explored (1) their cohesion as a latent LH factor and (2) the relationships between this factor and various predictor and criterion variables (e.g., *Chua et al., 2017*; *Dunkel & Decker, 2010*; *Figueredo et al., 2004*; *Figueredo et al., 2005*; *Figueredo et al., 2007*; *Figueredo et al., 2015*; but see (*Richardson et al., 2017*) for an alternative view). This research tradition uses self-report measures of constructs such as the major dimensions of personality, attachment style, positive and negative affect, paranoia, and general trust. A slower LHS is associated with higher levels of conscientiousness, agreeableness and emotional stability; a secure attachment style; more positive affect; less negative affect; greater religiosity; and a more prosocial orientation toward conspecifics.

A still largely unexplored set of questions concerns the relationships between LHS, as a high-level psychological construct, and quotidian behavior measured via direct observation. In other words, little has been done to describe the *ethology* of human LHS variation (but see *Sherman, Figueredo & Funder, 2013*). For example, do faster and slower LH strategists differ in how they allocate their time across activities, or in how they conduct face-to-face interactions with friends, romantic partners, or colleagues? These questions are important for three reasons. First, the understanding of individual differences, from any theoretical perspective, can be greatly enriched by studying naturally occurring behavior in addition to self-report ratings and laboratory task responses (*Funder, 2001*; *Furr, 2009*). Second, answers to these questions will suggest the extent to which human LHS is detectable to intimate, or even casual, observers, and they may provide a foundation for hypotheses about adaptations for detecting LHS in others. Given that personality variation is one manifestation of LHS variation (*Figueredo et al., 2007*; *Manson, 2017a*; *Rushton, 1985*), and that personality is somewhat detectable from casual observation (*Funder, 2012*), this is a potentially fruitful line of inquiry. Third, an ethology of LHS will help bridge the gap between traditional evolutionary psychology and differential evolutionary psychology (see *Buss, 2009*). The former emphasizes species-typical, semi-autonomous, domain-specific modules (*Tooby & Cosmides, 1990*), whereas the latter emphasizes evolutionary processes (e.g., balancing selection, mutation-selection balance) that generate and maintain individual variation in domain-specific reaction norms (e.g., *Nettle & Penke, 2010*;

*Verweij et al., 2012*). According to LH theory, these reaction norms governing different domains are inter-correlated rather than independent (*Del Giudice, 2014*; *Figueredo, Cabeza de Baca & Woodley, 2013*). Individuals differ in their "settings" of LH-related species-typical modules (e.g., risk-proneness), and these differences can be probed by experiments (e.g., *Griskevicius et al., 2011*). However, natural selection has likely acted on individual differences in situation selection (*Buss, 1987*; *De Vries et al., 2016*; *Nettle & Penke, 2010*)—for example, people differ in their self-chosen exposure to situations in which risk-taking is salient. Merely knowing how a person responds to a laboratory manipulation reveals nothing about such variation. Furthermore, different situations afford the expression of different personality traits (*Reis, 2008*)—for example, it's much easier to express extraversion at a party than while writing a manuscript. These issues can only be explored by systematic observation of naturally occurring behavior, using theory-driven behavioral categorization.

## The present study

As a preliminary inquiry into the ethology of LHS variation, I address a simple question: how well does a widely-used psychometric LHS instrument, the Arizona Life History Battery (ALHB: *Figueredo, 2007*) predict the everyday behavior of a sample of US college students? The ALHB, or its short form the Mini-K, has been used in many studies (*Figueredo et al., 2015*), but almost all of this work describes associations with other self-report instruments. A possible weakness of the ALHB/Mini-K is its undercoverage of aspects of extraversion (dominance striving, sensation-seeking) and openness to experience (imagination, unconventionality) that are theoretically expected (*Del Giudice, 2014*) to be associated with a faster LHS. In earlier analyses of the data set used in this paper (*Manson, 2017a*), self-reported ALHB scores were found to be only weakly (non-significantly) associated with other-reported LHS as indicated by similarity to the slow life history template (*Dunkel et al., 2015*; *Sherman, Figueredo & Funder, 2013*) of the California Adult Q-Sort (CAQ: *Block, 1978*). Analysis of this template indicates that it taps the fast LHS-associated facets of extraversion and openness (*Manson, 2017a*). Furthermore, the slow LHS template of the CAQ has been validated by correlating it with biodemographic LH indicators such as age at sexual debut and number of sex partners (*Dunkel et al., 2015*; see also *Kubinski, Chopik & Grimm, 2017*).

In the present study, participants' everyday behavior was sampled using the Electronically Activated Recorder (EAR: *Mehl et al., 2001*; *Mehl & Robbins, 2012*). The EAR unobtrusively collects periodic brief audio snippets of people's daily lives, using a portable recording device that participants wear attached to their clothes during their waking hours. Behavioral variables coded from EAR recordings (e.g., location and general activity) have shown substantial inter-rater reliability and within-participant temporal stability (*Mehl & Pennebaker, 2003*), and have provided novel insights into processes such as parent–child transmission of negative emotionality (*Slatcher & Trentacosta, 2012*) and the role of family environment in children's recovery from trauma (*Alisic et al., 2015*). Self-reported major personality dimensions are correlated in predictable ways with some EAR-measured

**Table 1  Coded behaviors.** Behaviors with kappa <.41 were excluded from further analysis.

| Behavior | Predicted association with ALHB | Cohen's kappa | Denominator ($n_i$ in binomial regression models) |
|---|---|---|---|
| Sleeping between 06:00 and 18:00 | Negative | .76 | Clips recorded 06:00–18:00 |
| Social interaction (including by phone, Skype, etc.) | Positive | .85 | Clips awake |
| Interactions including >1 interlocutor | Positive | .76 | Clips in social interaction |
| Attending class | Positive | .88 | Weekdays 08:00–17:00 |
| Watching TV, movie or video | Negative | .73 | Clips awake |
| Videogame playing | Negative | .52 | Clips awake |
| At amusement venue (bar, concert, athletic event) | Negative | .84 | Clips awake |
| Religious service or study group | Positive | .65 | Clips awake |
| Volunteer service | Positive | .61 | Clips awake |
| Arguing (disagreement accompanied by anger) | Negative | .49 | Clips in which P speaks |
| Talk about future plans (>1 year from present) | Positive | .80 | Clips in which P speaks |
| Talk about past experiences (>1 year ago) | Positive | .30 | N.A. |
| Talk approvingly about alcohol or recreational drug use | Negative | .60 | Clips in which P speaks |
| Talk about kin | Positive | .39 | N.A. |
| Talk to kin | Positive | .17 | N.A. |
| General complaining | Negative | .39 | N.A. |
| Anti-authority talk | Negative | .11 | N.A. |
| Sighing | Negative | .26 | N.A. |

variables (e.g., extraversion negatively correlated with proportion of time spent alone; *Mehl, Gosling & Pennebaker, 2006*).

Table 1 lists the behaviors measured in the present study, and their predicted associations with slow life history strategy as measured by the ALHB, based on findings from psychometric studies (*Brumbach, Figueredo & Ellis, 2009*; *Figueredo et al., 2007*; *Figueredo et al., 2005*; *Figueredo et al., 2014a*; *Sherman, Figueredo & Funder, 2013*). College students pursuing a slower LHS were predicted to engage more frequently in behavior indicative of (1) future time perspective, the ability to learn from past experience, and investment in embodied capital (*Kaplan et al., 2000*) (*class attendance, talking about long-term plans* and *talking about past experiences*), (2) altruistic dispositions toward kin and the wider community (*talking to kin; talking about kin; volunteering*); and (3) religiosity (*attending religious services and study groups*). A slower LHS was predicted to be associated with lower rates of (1) entertainment consumption (*TV and movie watching; videogame playing; being at amusement venues* such as bars, concerts, and sports events), (2) *talk endorsing alcohol or recreational drug use*, (3) indicators of negative affect (*general complaining; sighing;* see *Robbins et al., 2011*; *Roth, 2005*), and (4) indicators of antagonistic social schemata (*Patch & Figueredo, 2017*) or an alienated social orientation (*arguing; complaining about authority figures*). A final prediction, that a slower LHS would be negatively related to *amount of daytime sleeping*, requires additional explanation. Research suggests that LHS is related to chronotype (morningness/eveningness, *Adan et al., 2012*), such that people pursuing a slower LHS are more active in the morning, whereas those pursuing a faster LHS are more active in the evening (*Ponzi et al., 2015*). Because the audio recording protocol in the

present study included an 00:00–06:00 blackout period (see Materials and Methods), and because participants rarely went to sleep before the blackout period began (see Results), sleeping time mostly consisted of sleeping between 06:00 and 18:00. Sleep time during this time interval (corresponding to late rising and afternoon napping) was isolated for analysis, and I tested the prediction that it would be negatively associated with a slower LHS.

Finally, because the ALHB/Mini-K is strongly positively related to extraversion (*Figueredo et al., 2014a*; *Gladden, Figueredo & Jacobs, 2009*; *Manson, 2015*; *Strouts, Brase & Dillon, 2016*), including among the present study's participants (*Manson, 2017a*), ALHB scores were predicted to be positively associated with two indicators of sociality (*engagement in social interaction*; *proportion of social interactions that involve more than one interlocutor*).

## MATERIALS AND METHODS

### Ethics statement

All procedures described here were approved by UCLA's Institutional Review Board (Approval #12-001128-AM-00001). Informed consent was obtained from all participants in accordance with the terms of this approval.

### Relation to previous research

The dataset used in the present study is the same as that used by *Manson (2017a)*, *Manson (2017b)* and *Manson & Robbins (2017)*, but the analyses reported here are novel.

### Participants

Ninety-two students (55.4% female, $M \pm SD = 20.0 \pm 3.1$ years old), enrolled at the University of California, Los Angeles, were recruited via posted flyers and classroom announcements to participate in a study bearing the public title "Audio Sampling of Daily Life." Based on self-reported ethnic identity, 53.2% of the sample was Asian or Asian-American, 16.3% White, 12.0% Latino/a, 3.2% Middle Eastern, and 15.3% mixed or "other." Although unrepresentative of college-aged Americans generally, the sample's ethnic composition was fairly representative of UCLA's undergraduate student body. Data were collected between October 2013 and January 2015.

### Procedure

#### Audio sampling

Recordings were made using the Electronically Activated Recorder (EAR: *Mehl et al., 2001*; *Mehl, Robbins & Deters, 2012*). Instantiated as an app for the iPod Touch, the EAR generates periodic brief audio recordings (in this study, a 30-sec clip every 12.5 min) as participants go about their daily lives. Participants know the general sampling pattern, but not whether the app is recording at any particular time. In the present study, participants were instructed to wear the iPod clipped to their belt or waistline whenever possible during their waking hours for a 72-hour period. Recording could begin on any weekday. No recordings were made between midnight and 0600. Participants were also instructed to keep an hourly diary briefly describing their general activities and whether or not they were wearing the iPod during that entire hour. Upon returning the iPod, participants

were given the opportunity to privately review their audio clips and to delete any they wished, before researchers listened to them. Participants received a $50 Amazon gift card as compensation. They were not informed of the specific hypotheses motivating the study until after all participants had completed participation.

***Self-report measures***

After returning the iPod and before receiving their compensation, participants were asked to complete, at their convenience, an online battery of self-report instruments, including the ALHB (*Figueredo, 2007*), a 199-item instrument consisting of eight scales drawn from various original sources (*Barrera Jr, Sandler & Ramsay, 1981*; *Brennan, Clark & Shaver, 1998*; *Brim et al., 2000*). Seven of the scales measure distinct aspects of life history strategy: insight, planning and control; relationships with biological parents; family contact and support; friends contact and support; general altruism; religiosity; and experiences in close relationships. The eighth ALHB scale, the Mini-*K*, contains 20 items tapping general features of all seven LHS facets. Because the sample was drawn from an undergraduate population, I deleted from analyses the eight items of the ALHB's altruism toward children subscale. The eight ALHB indicators converge on a single multivariate latent construct, the *K*-Factor (*Gladden, Figueredo & Jacobs, 2009*; *Wenner et al., 2013*), which can be estimated as the mean of the *z*-scores of the seven scales excluding the Mini-*K* (*Figueredo, 2007*). Higher *K*-Factor scores indicate a slower LHS.

## Data preparation and analysis
### EAR behavior coding

Each audio clip was coded by a trained research assistant with respect to 31 behavioral variables, including two that were measured to assess effects of the EAR method itself on behavior: *compliance*, i.e., whether the iPod was on the participant's person, or at least close enough to generate valid recordings; and *talk about participating in the study*, as a measure of obtrusiveness. Results indicating acceptably high compliance and low obtrusiveness are presented by *Manson & Robbins (2017)*. Of the remaining 29 behaviors, 18 are relevant to the present study (Table 1).

Each participant's clips were coded by one coder. Inter-rater reliability was assessed by assigning the 11 research assistants to code a test set of 110 clips containing at least one exemplar of every coded behavior category. Research assistants coded these clips independently, and inter-rater reliability was measured with Cohen's kappa. A kappa value of .41 (moderate agreement according to *Landis & Koch, 1977*) was pre-selected as the cut-off below which variables would be excluded from further analysis.

## Modeling relationships between predictor variables and behavior frequencies

To assess relationships between the ALHB and behavior frequencies, I carried out Bayesian multi-level aggregated binomial regressions, using inferential procedures and notation described by *McElreath (2015)*. Previous EAR-based research (e.g., *Baddeley, Pennebaker & Beevers, 2013*; *Mehl, Gosling & Pennebaker, 2006*) has scored behavioral variables as proportions, e.g., the number of a participant's audio clips containing dyadic interactions

divided by the total number of that participant's clips containing interactions (dyadic plus larger conversational groups). However, converting count data to proportions or percentages discards the information contained in the magnitude of the denominator; e.g., the observation that one participant's multi-interlocutor social interactions comprised 10 out of 20 clips containing any social interaction is less reliable (contains less information) than the observation that another participant's multi-interlocutor interactions comprised 60 out of 120 clips containing any interaction. A binomial regression procedure models the observed (count) form of the data as a binary outcome (here, whether a particular behavior does or does not occur in a particular audio clip). A multi-level binomial regression, using participants as clusters, estimates both an overall intercept (here, the probability of a behavior occurring in a clip when the predictor variable[s] is at its mean value) and a specific intercept for each participant. Thus, the model takes into account (1) unmeasured differences between participants that are related to the occurrence of the designated behavior, (2) the additional uncertainty introduced by rarely occurring behaviors, and (3) skewed distributions of behavior frequencies. Estimates of participant-specific intercepts that are farther from the overall intercept, and/or based on a smaller denominator (i.e., the number of clips in which the behavior could possibly have occurred), are "shrunk" farther toward the overall intercept. These intercepts are then incorporated into a linear model of the relationship between the predictor (here, the LHS or personality measure) and the probability of a behavior occurring during a given clip. A logit link maps this probability (which must lie between 0 and 1) onto a linear model (in which the outcome variable is expressed as log-odds). Rather than a $p$-value, each Bayesian model yields a posterior distribution of parameter value combinations, i.e., the relative probability of having generated the observed data associated with each combination of intercept, "slope" (i.e., the relationship between the predictor variable and the probability of engaging in that behavior), and standard deviation of the outcome variable. As an example, the relationship between ALHB score and the frequency of multi-interlocutor social interactions was modeled as follows:

$$\text{Count-of-multi-interlocutor-interactions}_i \sim \text{Binomial}\,(n_i, p_i) \tag{1}$$

$$\text{logit}(p_i) = \alpha + \alpha_{\text{PARTICIPANT}[i]} + (\beta * \text{ALHB}_i) \tag{2}$$

$$\alpha \sim \text{Normal}(0, 2) \tag{3}$$

$$\beta \sim \text{Normal}\,(0, 2) \tag{4}$$

$$\alpha_{\text{PARTICIPANT}[i]} \sim \text{Normal}\,(0, \sigma_{\text{PARTICIPANT}}) \tag{5}$$

$$\alpha_{\text{PARTICIPANT}} \sim \text{HalfCauchy}\,(0, 1) \tag{6}$$

Line (1), the likelihood, states that the number of clips containing multi-interlocutor interactions involving participant $i$ is distributed binomially, with $n_i$ as the total number of $i$'s clips containing any social interactions, and $p_i$ as the probability that a clip containing social interaction by participant $i$ contains a multi-interlocutor interaction. Line (2) is the logit link function, incorporating the overall intercept, the participant-specific intercept offset, and the slope ($\beta$). Lines (3) through (6) are the prior distributions (described as

**Table 2  Observed frequencies of behaviors.** For definitions of possible clips for each behavior, see Table 1

| Behavior | Proportion of possible clips in which behavior was observed | | Proportion of participants observed to engage in behavior |
|---|---|---|---|
| | Mean ± SD | Range | |
| Sleeping between 06:00 and 18:00 | .278 ± .130 | .000–.744 | .99 |
| Social interaction (including by phone, Skype, etc.) | .323 ± .166 | .022–.731 | 1.00 |
| Interactions including >1 interlocutor | .391 ± .182 | .000–.800 | .98 |
| Attending class | .196 ± .143 | .000–.702 | .86 |
| Watching TV, movie or video | .076 ± .096 | .000–.433 | .88 |
| Videogame playing | .020 ± .059 | .000–.353 | .31 |
| At amusement venue (bar, concert, athletic event) | .013 ± .030 | .000–.137 | .20 |
| Religious service or study group | .007 ± .025 | .000–.157 | .13 |
| Volunteer service | .008 ± .026 | .000–.151 | .14 |
| Arguing (disagreement accompanied by anger) | .013 ± .034 | .000–.250 | .31 |
| Talk about future plans  (>1 year from present) | .015 ± .025 | .000–.105 | .41 |
| Talk approvingly about alcohol or recreational drug use | .005 ± .015 | .000–.103 | .15 |

mean, standard deviation) for the overall intercept, the slope, the participant-specific intercept, and the participant-specific standard deviation.

Before modeling relationships between ALHB scores and behavior frequencies, I modeled relationships between behavior frequencies and (1) whether or not a clip was recorded during a weekend (17:00 Friday to 24:00 Sunday, or during a school holiday) and (2) the participant's sex. These were treated as dummy variables, with weekend and female coded as 1, while weekday and male were coded as 0. *Class attendance*, alone among all analyzed behaviors, was assumed to be possible only during weekdays (see Table 1). Weekend is a nuisance variable in the context of the present study. Sex, however, is a theoretically noteworthy variable, in that women generally pursue slower life history strategies than men (*Figueredo et al., 2005*; *Kubinski, Chopik & Grimm, 2017*). Because of the rarity (see Table 2) of two of the recorded behaviors, *volunteering* and *religious service/study*, and consequent problems with model performance, clips containing either of these behaviors were summed into a single variable, predicted to be positively associated with the ALHB.

For 61 of the 91 participants (67%), $n_i$ (the denominator, i.e., the number of clips in which the behavior could have been observed) comprised all clips in which the participant was compliant and awake for the behaviors *social interaction, presence at amusement venues, TV and movie watching, videogame playing* and the aggregated *religious services/volunteering* category. For the remaining 30 participants, $n_i$ varied across these behavior categories because one or more of these behaviors could not be coded in one or more clips (e.g., the participant was at a bar in which the music was so loud that it was impossible to determine whether he or she was engaged in social interaction). The maximum percentage of a participant's awake/compliant clips that was subtracted from the total for such an adjustment was 6.6%. See Data S1 for more details.

Each model generated by these procedures was then compared, using the Widely Applicable Information Criterion, to a null model (i.e., a model of the same behavior containing no predictor variable, only the intercept). Information criteria reward models for explaining more variance in the dependent variable, but penalize models for additional parameters, thus reducing overfitting. The Akaike weights of the compared models sum to 1. A model's weight is an estimate of the probability that the model will make the best predictions on new data, conditional on the set of models considered. For each behavior, all models with Akaike weights greater than that of the null model were then compared, as a single set of models, to each other as well as to the null model. Interpretations took into account both (1) the distribution of a model's estimated value of $\beta$ (i.e., the extent to which its credible interval overlapped zero) and (2) the model's Akaike weight relative to other models predicting that behavior.

Models were fitted using the Markov chain Monte Carlo estimation engine Stan, via the R Rethinking package (*McElreath, 2016*; *R Core Team, 2016*; *Stan Development Team, 2017*).

Because I examined the relationships of many behaviors to the ALHB, correction for multiple tests is necessary. For those behaviors that, based on the procedures described above, appeared to be predicted by the ALHB, I randomly re-assigned the ALHB scores (but no other variables) among the participants and re-ran the models 100 times. I then compared the estimated slopes and Akaike weights of the models based on the real data to the distributions of slopes and Akaike weights generated from these resampled data sets.

## RESULTS

### Descriptive statistics

As reported by *Manson (2017a)*, participants deleted an average of 1.1% of their audio clips (SD = 2.4%, median = 0, range = 0–15.0%). The mean number of compliant clips per participant was 232 (SD = 25.2, range = 148–261). For 43 of the 91 participants (47.3%), at least one clip was recorded during a weekend. Of 21,112 compliant clips, 5679 (26.9%) were recorded during weekends.

Table 1 shows Cohen's kappa for each of the 18 behaviors analyzed. Six behaviors (*sighing, talking to kin*, and four talk topics) were excluded from further analysis because of inadequate ($\kappa < .41$) inter-rater reliability. Table 1 also describes, for each behavior, the relevant denominator, i.e., the set of audio clips in which the behavior could possibly have been observed ($n_i$ in the binomial models). Table 2 presents data on the distributions of observed frequencies of each of the 12 behaviors with adequate inter-rater reliability. Five of the behaviors were observed at least once in >80% of participants, whereas three behaviors (*volunteering, attending religious services or study*, and *talk endorsing alcohol or recreational drug use*) were observed in <20% of participants.

Participants rarely went to sleep before the overnight non-recording period began at 24:00. Sixty-seven of 91 (73.6%) participants were never observed to do so. Among 243 observation days in which participant compliance continued until 24:00, participants were still awake at 24:00 on 209 (86.0%) days. One participant chose to delete all his early
morning clips, reducing the sample size to 90 for analyses of daytime sleeping. From these 90 participants, of 4,306 audio clips in which sleeping was recorded, only 338 (7.8%) were recorded between 18:00 and 24:00.

Four participants, who participated during summer 2014, were dropped from the analysis of predictors of classroom attendance, because inspection of their audio clips and event diaries indicated that there were not enrolled in any classes during the recording period.

As reported in by *Manson (2017a)*, the ALHB's internal reliability (i.e., across its seven scales) was .68.

## Model tests

Seven behaviors were more frequent during weekends than weekdays. This was evident both in the estimates of $\beta$ (positive, with the 95% credible interval not including zero), and in comparisons of models using weekend as a predictor to null models of the same behaviors (the former having 100% of the Akaike weight in all seven cases). These seven behaviors were *daytime sleeping*, *social interaction*, *proportion of social interactions that involve more than one interlocutor*, *presence at amusement venues*, *TV and movie watching*, *videogame playing*, and the aggregated *religious services/volunteering* category. For these behaviors, models of their associations with ALHB were compared to models including weekend as the only predictor (hereafter, weekend-only models). Only one behavior showed a sex difference, such that the 95% CI of the estimate of $\beta$ did not include zero, and the model with sex as a predictor had more Akaike weight than the null model. This behavior was *videogame playing*, which was engaged in more by men than by women ($\beta = -2.76$, 93% CI $= [-4.23, -1.28]$, Akaike weight of predictor model = 84%). The model of the relationship between the ALHB and *videogame playing* were compared to a model in which weekend and sex were the only predictors.

Table 3 shows the results of tests of models using ALHB as a predictor of behavior frequencies. Only one behavior, *presence at amusement venues*, was associated with ALHB scores (Akaike weight compared to the weekend-only model: 1.00). This association was in the opposite of the predicted direction: people whose ALHB score indicated a slower life history strategy spent more time at amusement venues than those whose ALHB score indicated a faster life history. However, random reassignment of ALHB scores among participants casts doubt on the strength of the relationship. Of the 100 runs of the model with randomly reassigned ALHB scores, six generated positive relationships between ALHB and *presence at amusement venues* with an Akaike weight of 1.00 compared to the intercept-only model (median estimate of $\beta$: 1.47). Five other runs generated negative relationships between ALHB and *presence at amusement venues* with an Akaike weight of 1.00 compared to the intercept-only model (median estimate of $\beta$: $-1.63$).

## DISCUSSION

The Arizona Life History Battery (*Figueredo, 2007*), and particularly its 20-item short form the Mini-K, have been widely used for the psychometric assessment of human LHS and have been validated by comparison with other self-report measures (*Figueredo et al., 2014a*). I

**Table 3  Results of Bayesian aggregated binomial regression analyses of associations between ALHB and behaviors.** Each row represents one model. Intercepts can be converted to estimated proportions by using the inverse link function logistic. Akaike weights are in comparison to appropriate null model (weekend as the only predictor, weekend plus sex as predictors, intercept only) as indicated in the left-hand column. Estimated $\beta$ [95% CI] of sex as a predictor of videogame playing: $-2.90$ [$-4.40$, $-1.49$].

| Behavior | | Predictor $\beta$(95% credible interval) | | Akaike weight |
|---|---|---|---|---|
| | $\alpha \pm SD$ | Weekend | ALHB | ALHB + weekend compared to weekend only |
| Daytime sleeping | $-1.35 \pm .07$ | 1.97 [.95, 1.19] | $-.18$ [$-.42$, .07] | .44 |
| Social interaction | $-1.05 \pm .10$ | .62 [.51, .74] | .19 [$-.11$, .52] | .55 |
| >1 interlocutor | $-.69 \pm .11$ | .66 [.46, .86] | .25 [$-.10$, .60] | .51 |
| Amusement venue | $-8.32 \pm .66$ | 1.16 [.68, 1.63] | 1.46 [$-.22$, 3.02] | 1.00 |
| Religious service + volunteer | $-7.95 \pm .61$ | 1.45 [.84, 2.05] | .56 [$-.91$, 2.06] | .00 |
| TV, movie, and video watching | $-3.50 \pm .17$ | .78 [.58, .98] | $-.06$ [$-.65$, .51] | .38 |
| | | | | ALHB + weekend + sex compared to weekend + sex only |
| Videogame playing | $-6.09 \pm .53$ | 1.70 [1.26, 2.16] | $-.06$ [$-1.38$, 1.24] | .00 |
| | | | | ALHB compared to intercept only |
| Attending class | $-1.65 \pm .13$ | N.A. | .13 [$-.29$, .54] | .50 |
| Arguing | $-4.92 \pm .26$ | N.A. | $-.61$ [$-1.31$, .06] | .05 |
| Talk about future plans | $-4.49 \pm .21$ | N.A. | $-.21$ [$-.80$, .38] | .06 |
| Talk endorsing alcohol or drugs | $-6.21 \pm .57$ | N.A. | 0.10 [$-1.07$, 1.17] | .00 |

Notes.
Videogame femaleness beta: $-2.90$ [$-4.40$, $-1.49$].

examined whether ALHB scores were associated, in theoretically deduced directions, with the frequencies of 12 everyday behaviors of California college students. Results failed to show any power for the ALHB in predicting the frequencies of these behaviors.

Whether these findings undermine confidence in the validity of the ALHB is unclear. It is possible to argue that frequencies of some of the behaviors would not necessarily be expected to reflect LHS variation. For example, because of the proliferation of smartphones and tablets, class attendance by college students may suffer from diminished validity as an indicator of investment in embodied capital. Audio recordings provide no information about whether a participant in a classroom is paying attention to what's happening in the classroom, as distinct from texting, checking social media, watching video, etc. In contrast, *Mehl, Gosling & Pennebaker (2006)*, who collected EAR data on college students in the early 2000s, found that classroom attendance was predicted by self-reported Big Five conscientiousness. As another example, I used daytime sleeping as an indicator of chronotype and hence an indirect indicator of LHS. However, a more valid measure of chronotype would incorporate participants' sleep onset times, which were unknown on 86% of observation days because of the ethical necessity to include an overnight blackout period in the EAR protocol. Furthermore, six behaviors showed inadequate inter-rater reliability and were therefore not available for analysis. Nevertheless, it is striking that none of the hypothesized behavioral indicators of LHS were associated in the predicted direction with ALHB scores. The ALHB's undercoverage of aspects of extraversion associated with a faster LHS (excitement-seeking, dominance striving; see *Del Giudice, 2014*; *Manson,*

*2017a*) may explain why ALHB scores were not associated as predicted with some of the behaviors listed in Table 1 (e.g., endorsing alcohol or drug use; presence at amusement venues; arguing). It is more surprising that ALHB scores were not associated with behaviors indicative of high extraversion (e.g., social interaction).

The study participants were US college students, as were the participants in most of the studies that have validated the ALHB and Mini-K with respect to other self-report instruments (*Figueredo et al., 2014a*). The EAR volunteers were in some respects unrepresentative of the student population from which they were drawn, being higher in conscientiousness and lower in emotionality than a sample of students from a course participant pool, at the same university, who did not volunteer for the EAR study (*Manson & Robbins, 2017*). However, sample unrepresentativeness is unlikely to account for the null results reported here, because the EAR volunteers did not differ from the participant pool comparison sample with respect to self-reported LHS (e.g., on the 7-point Mini-K, the EAR participants' mean $\pm$ SD was 5.23 $\pm$ .65, compared to 5.22 $\pm$ .72 [$N = 161$] for the participant pool sample).

A final limitation of the present study is that it examined associations between a stable individual characteristic (life history strategy) and quotidian behavior without accounting for the influence of situations on behavior, or the ways in which individual characteristics, situational characteristics, and behavior interact (*De Vries et al., 2016*; *Rauthmann, Sherman & Funder, 2015*; *Sherman, Nave & Funder, 2010*). These processes can be expected to diminish the strength of simple associations between self-reported individual traits and everyday behavior, and may account for the null findings of the present study as well as the paucity of strong associations between self-reported major dimensions of personality and quotidian behavior reported in the EAR study of *Mehl, Gosling & Pennebaker (2006)*. Ongoing analyses of a subset of the data used here (specifically, audio clips in which participants spoke) are addressing this issue by using other-rated scores on the Riverside Behavioral Q-Sort (*Furr, Wagerman & Funder, 2010*) and the Riverside Situational Q-Sort (*Wagerman & Funder, 2009*) as a basis for hypothesis testing.

In addition to its substantive findings, the present study illustrates the usefulness of count models (here, binomial regression models) when behavioral observation protocols generate count variables. The common practice of converting counts to proportions unnecessarily discards information helpful to inference (*McElreath, 2015*). The present study also followed the recent recommendation of some statisticians (*McElreath, 2015*; *McShane et al., 2017*) to use alternatives to null hypothesis testing.

## CONCLUSIONS

This study failed to find relationships between the Arizona Life History Battery and frequencies of 12 of everyday behaviors in US college students. Additional research is necessary to explore the possibility of subtler relationships between psychometrically-assessed life history strategy and quotidian behavior.

### Funding

This research was supported by a Faculty Research Grant from the UCLA Academic Senate. The funders had no role in study design, data collection and analysis, decision to publish, or preparation of the manuscript.

### Grant Disclosures

The following grant information was disclosed by the author:
UCLA Academic Senate.

### Competing Interests

The author declares there are no competing interests.

### Author Contributions

- Joseph H Manson conceived and designed the experiments, performed the experiments, analyzed the data, contributed reagents/materials/analysis tools, prepared figures and/or tables, authored or reviewed drafts of the paper, approved the final draft.

### Human Ethics

The following information was supplied relating to ethical approvals (i.e., approving body and any reference numbers):

All procedures described here were approved by UCLA's Institutional Review Board (Approval #12-001128-AM-00001). Informed consent was obtained from all participants in accordance with the terms of this approval.

### Data Availability

Manson, Joseph (2018): Data_JHManson_Life-history-strategy-and-daily-behavior.csv. figshare: https://figshare.com/articles/_/5709850.

### Supplemental Information

Supplemental information for this article can be found online at http://dx.doi.org/10.7717/peerj.4866#supplemental-information.

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
