# Peer review of "Associations between psychometrically assessed life history strategy and daily behavior: data from the Electronically Activated Recorder (EAR)"

_PeerJ, doi:10.7717/peerj.4866_

## Round 0.1 · original submission · Major Revisions

One of the reviewers offered several substantial critiques, and they appear to have substantial merit. Please address these as comprehensively as possible in your revised manuscript and rebuttal.

·

Basic reporting

The standard of reporting is generally very full and clear. However, I do confess that I find the inclusion of the HEXACO analyses confusing. I can grasp the idea of a study to see whether life history strategy constructs predict quotidian behaviours. Then I can grasp that there are three different measures of life history strategy, that these do not correlate well with one another, and that hence you need to look at each different life history strategy measure in relation to each behaviour. But then introducing six *other* constructs, that seem to me to be related to a different set of questions, and to relate each of those to every behaviour – it causes huge issues of multiple testing, and to my mind it starts to make the whole paper incoherent. Are the HEXACO dimensions supposed to be sub-constructs of life history strategy (this is how they are often discussed)? Or are they alternative predictors when life history strategy does not work (as table 5 implies). This just doesn’t make any clear sense to me, and it overlaps with other papers already published by the author. I would advise taking the HEXACO stuff right out in order to be clearer about the rationale and interpretation of the life history strategy material.

Experimental design

I really want to commend the author on his interest in quotidian behaviour, and his willingness to go the long hard road of a behavioural study using the EAR. Relating quotidian behaviour to personality or psychometric constructs as usually used in psychology is a great objective and this data set is a non-trivial achievement. I also want to commend the use of the Bayesian stats, and going right down to the raw behavioural outcomes rather than aggregating.

Despite these strengths, I have some major concerns about the design.

1. There seems to me to be a huge problem of multiple testing, which the Bayesian stats do nothing to address. You get enough behaviours, and test the probability of occurrence of each of them with each of several psychometric predictors, and sure enough, you find some positive associations stronger than chance. They are not in general the ones you predicted, and not the same ones for the three measures of life-history strategy, and there are not very many of them. We need some sense of how many associations you would get by chance with that degree of multiple testing anyway. Relatedly, no very informative measure of effect size, like R2 or proportionate reduction in variance, is offered, so we don’t know if knowing life-history strategy dramatically or only marginally improves our ability to predict behaviour. In effect, we need a null model and a sense of to what extent life history strategy measures allow better prediction than the null model. There are several ways to implement this within the author’s framework. One thing I would suggest is to scramble the dataset. That is, randomly re-assign all the life history scores to different people. Then rerun the analyses. Indeed the beauty of working in R is that you can do this many times. I think we need an overall sense of how much better we do with real measures of life-history strategy than with random data. I’d like to see this quantified – I think the answer may well be not much. This often happens in psychometric research. People predict lots of associations, find a few of them, and then present the results as partial support for the predictive framework. But it is not at all obvious that it is even partial support.

2. Another problem that strikes me is that the life history strategy measures are of two categorically distinct kinds. The AHLB is a separate source of information from the EAR recordings and diaries. Thus, correlations between it and the recordings and diaries do tell us something. The CAQ measures, as I understand them, were made by raters *using* the EAR recordings and diaries. Thus, they are not independent of those recordings and diaries. Any correlation between the recordings/diaries and the CAQ thus really just tell us how the raters did their job – they used information in the EAR data set to produce the CAQ. Correlations should be expected. In fact, it’s odd they are so weak. For me this is really problematic. Basically you are using a human-created qualitative summary of a set of data to try to predict what is in that data – so what does that tell us about the actual participants? I don’t have a solution to this, except that for me the CAQ stuff suffers from a kind of circularity, and only the ALHB stuff is telling us anything about the relationship between the life history construct and quotidian behaviour.

Validity of the findings

I think you can tell from my comments under experimental design what my concerns about the validity of the findings are. I don’t quite know what implication the author intends. For example, there are two measures of life-history strategy here that are supposed to be measuring the same thing: the CAQ overall, and the ALHB, and they are basically uncorrelated (I really had to dig to find this information). Not only that but they have different patterns of association with other constructs like the HEXACO. What does this mean for the whole construct? To my mind it renders it problematic, and yet there is no critical discussion of the utility of the construct.

Then there is the fact that most of the predicted associations don’t materialise with any of the measures, and no association appears with all of them (and see experimental design above for my concern about circularity with the CAQ score anyway). There is even one in the opposite direction. So is it not time to start doubting the utility of the slow life history construct? What would it take for the author to conclude that the construct was not useful in predicting behaviour, or is not really measuring a thing?

In short, I don’t quite know what hypothesis is under test, and hence don’t know what answer comes back. Maybe the author could consider posing himself a more sharply defined question, such as: ‘How well does the Arizona Life History Battery score allow us to predict quotidian behaviour?’. Then, we could come up with clear quantitative answer: we can reduce our uncertainty by x% by knowing the AHLB score compared to not knowing the AHLB score. There are a couple of ways you could do this. 1. The data randomization strategy described in Experimental Design; or 2. Calculating something like the proportional reduction in variance (PRV) due to ALHB for each behaviour, and then averaging this across behaviours (see e.g. Nakagawa, S. & Schielzeth, H. 2013. A general and simple method for obtaining R2 from generalized linear mixed-effects models. Methods Ecol. Evol. 4, 133–142).

Additional comments

The author knows me, I like his work, and so let me say I am sorry for giving him a challenging review. He will no doubt come back at me if he disagrees with my characterization. I have gone out of my way to commend his interest in measuring quotidian behaviour, and the use of the EAR method in particular. It’s a useful data set, but I think the paper would have more impact if conceptually tidied up, streamlined, and answering a clearer, sharper non-circular problem.

·

Basic reporting

The paper is clear and well-written.

There are a few spots that should be corrected:

-“enrolled at” is repeated twice in the participants subsection.
-“vary” should be “very” under “California Adult Q-sort (CAQ) life history strategy scoring” section

Experimental design

The design is rigorous with multiple measures to assess LHS. The Methods are described in detail.

Validity of the findings

The psychometric approach to assessing LHS has been mostly deficient in looking at observations of behavior (ethograms) rather than only reported behavior. This paper is helpful filling that gap and doing so with some diversity of measures of LH strategy.

Additional comments

My bottom line is that I think the paper is excellent and should be accepted for publication—without much revision. It makes an important contribution in the study of how measures of LHS relate to daily behaviors. The psychometric approach to LHS has been deficient in looking at observations of behavior (ethograms) rather than only reported behavior. This paper is helpful filling that gap and doing so with some diversity of measures of LH strategy.

Paul R. Gladden, Ph.D.
Interim Department Chair
Associate Professor of Psychology
Department of Psychology and Criminal Justice
Middle Georgia State University
Macon, GA
[email protected]

---

## Round 0.2 · accepted · Accept

This revision represents a major improvement. The findings and major message are now clear.

# ·

Basic reporting

The standard of reporting is very high.

Experimental design

The study is now well-designed, once the various extra measures of the earlier version have been cleared away.

Validity of the findings

I think the findings are valid and we can have some confidence in them, albeit that they are mainly neutral associations.

Additional comments

I'd like to congratulate the author on the revisions. It took some wielding of the scalpel, but I think the resulting manuscript is much clearer and sounder.